# A General Computational Framework to Measure the Expressiveness of Complex Networks using a Tight Upper Bound of Linear Regions

## Abstract

The expressiveness of deep neural network (DNN) is a perspective to understand the surprising performance of DNN. The number of linear regions, i.e. pieces that a piece-wise-linear function represented by a DNN, is generally used to measure the expressiveness. And the upper bound of regions number partitioned by a rectifier network, instead of the number itself, is a more practical measurement of expressiveness of a rectifier DNN. In this work, we propose a new and tighter upper bound of regions number. Inspired by the proof of this upper bound and the framework of matrix computation in Hinz & Van de Geer (2019), we propose a general computational approach to compute a tight upper bound of regions number for theoretically any network structures (e.g. DNN with all kind of skip connections and residual structures). Our experiments show our upper bound is tighter than existing ones, and explain why skip connections and residual structures can improve network performance.

## 1 Introduction

Deep nerual network (DNN) (LeCun et al., 2015) has obtained great success in many fields such as computer vision, speech recognition and neural language process (Krizhevsky et al., 2012; Hinton et al., 2012; Devlin et al., 2018; Goodfellow et al., 2014). However, it has not been completely understood why DNNs can perform well with satisfying generalization on different tasks. Expressiveness is one perspective used to address this open question. More specifically, one can theoretically study expressiveness of DNNs using approximation theory (Cybenko, 1989; Hornik et al., 1989; Hanin, 2019; Mhaskar & Poggio, 2016; Arora et al., 2016), or measure the expressiveness of a DNN. While sigmoid or tanh functions are employed as the activation functions in early work of DNNs, rectified linear units (ReLU) or other piece-wise linear functions are more popular in nowadays. Yarotsky (2017) has proved that any DNN with piece-wise linear activation functions can be transformed to a DNN with ReLU. Thus, the study of expressiveness usually focuses on ReLU DNNs. It is known that a ReLU DNN represents a piece-wise linear (PWL) function, which can be regarded to have different linear transforms for each region. And with more regions the PWL function is more complex and has stronger expressive ability. Therefore, the number of linear regions is intuitively a meaningful measurement of expressiveness (Pascanu et al., 2013; Montufar et al., 2014; Raghu et al., 2017; Serra et al., 2018; Hinz & Van de Geer, 2019).

A direct measurement of linear regions number is difficult, if not impossible, and thus the upper bound of linear regions number is practically used as a figure of metrics to characterize the expressiveness. Inspired by the computational framework in (Hinz & Van de Geer, 2019), we improve the upper bound in Serra et al. (2018) for multilayer perceptrons (MLPs) and extend the framework to more complex networks. More importantly, we propose a general approach to construct a more accurate upper bound for almost any type of network. The contributions of this paper are listed as follows.

- Through a geometric analysis, we derive a recursive formula for $\gamma$, which is a key parameter to construct a tight upper bound. Employing a better initial value, we propose a tighter upper bound for deep fully-connected ReLU networks. In addition, the recursive formula provide a potential to further improve the upper bound given an improved initial value.

- Different from Hinz & Van de Geer (2019), we not only consider deep fully-connected ReLU networks, but also extend the computational framework to more widely used network architectures, such as skip connections, pooling layers and so on. With the extension, the upper bound of U-Net (Ronneberger et al., 2015) or other common networks can be computed. By comparing the upper bound of different networks, we show the relation between expressiveness of networks with or without special structures.

- Our experiments show that novel network structures enhance the upper bound in most cases. For cases in which the upper bound is almost not enhanced by novel network settings, we explain it by analysing the partition efficiency and the practical number of linear regions.

## 2 RELATED WORK AND PRELIMINARIES

### 2.1 RELATED WORK

There are literature on the linear regions number in the case of ReLU DNNs. Pascanu et al. (2013) compare the linear regions number of shallow networks by providing a lower bound. Montufar et al. (2014) give a simple but improved upper bound compared with Pascanu et al. (2013). Montúfar (2017) proposes a even tighter upper bound than Montufar et al. (2014). And Raghu et al. (2017) also prove a similar result which has the same order compared to Montúfar (2017). Later, Serra et al. (2018) propose a tighter upper bound and a method to count the practical number of linear regions. Furthermore, Serra & Ramalingam (2018); Hanin & Rolnick (2019a;b) explore the properties of the practical number of linear regions. Finally, Hinz & Van de Geer (2019) employ the form of matrix computation to erect a framework to compute the upper bound, which is a generalization of previous work (Montufar et al., 2014; Montúfar, 2017; Serra et al., 2018)

### 2.2 NOTATIONS, DEFINITIONS AND PROPERTIES

In this section, we will introduce some definitions and propositions. Since the main computational framework is inspired by Hinz & Van de Geer (2019), some notations and definitions are similar.

Let us assume a ReLU MLP has the form as follows.

$$f(\boldsymbol{x}) = \boldsymbol{W}^{(L)} \underbrace{\sigma(\boldsymbol{W}^{(L-1)} \cdots \sigma}_{(L-1)}(\boldsymbol{W}^{(1)}\boldsymbol{x} + \boldsymbol{b}^{(1)}) \cdots + \boldsymbol{b}^{(L-1)}) + \boldsymbol{b}^{(L)} \tag{1}$$

where $\boldsymbol{x} \in \mathbb{R}^{n_0}, \boldsymbol{W}^{(i)} \in \mathbb{R}^{n_i \times n_{i-1}}, \boldsymbol{b}^{(i)} \in \mathbb{R}^{n_i}$ and $\sigma(\boldsymbol{x}) = \max(\boldsymbol{x}, 0)$ denoting the ReLU function. $\boldsymbol{W}^{(i)}$ is the weights in the $i^{\text{th}}$ layer and $\boldsymbol{b}^{(i)}$ is the bias vector. $f(\boldsymbol{x})$ can also be written as:

$$h_0(\boldsymbol{x}) = \boldsymbol{x}, h_i(\boldsymbol{x}) = \sigma(\boldsymbol{W}^{(i)} h_{i-1}(\boldsymbol{x}) + \boldsymbol{b}^{(i)}), 1 \leq i < L, \tag{2}$$

$$f(\boldsymbol{x}) = h_L(\boldsymbol{x}) = \boldsymbol{W}^{(L)} h_{L-1}(\boldsymbol{x}) + \boldsymbol{b}^{(L)} \tag{3}$$

Firstly, we define the linear region in the following way.

**Definition 1.** *For a PWL function $f(\boldsymbol{x}) : \mathbb{R}^{n_0} \to \mathbb{R}^{n_L}$, we define $\mathbb{D}$ is a linear region, if $\mathbb{D}$ satisfies that: (a) $\mathbb{D}$ is connected; (b) $f$ is an affine function on $\mathbb{D}$; (c) Any $\mathbb{D}' \supsetneq \mathbb{D}$, $f$ is not affine on $\mathbb{D}'$.*

For a PWL function $f$, the domain can be partitioned into different linear regions. Let

$$\mathbb{P}(f) = \{\mathbb{D}_i | \mathbb{D}_i \text{ is a linear region of } f, \forall \mathbb{D}_i \neq \mathbb{D}_j, \mathbb{D}_i \cap \mathbb{D}_j = \emptyset\}$$

represent all the linear regions of $f$. We then define the activation pattern of ReLU DNNs as follows.

**Definition 2.** *For any $\boldsymbol{x} \in \mathbb{R}^{n_0}$, we define the activation pattern of $\boldsymbol{x}$ in $i^{th}$ layer $\boldsymbol{s}_{h_i}(\boldsymbol{x}) \in \{0, 1\}^{n_i}$ as follows.*

$$\boldsymbol{s}_{h_i}(\boldsymbol{x})_j = \begin{cases} 1, & \text{if } \boldsymbol{W}_{j,:}^{(i)} h_{i-1}(\boldsymbol{x}) + b_j^{(i)} > 0 \\ 0, & \text{if } \boldsymbol{W}_{j,:}^{(i)} h_{i-1}(\boldsymbol{x}) + b_j^{(i)} \leq 0 \end{cases}, \text{ for } i \in \{1, 2, \ldots, L-1\}, j \in \{1, 2, \ldots, n_i\},$$

*where $\boldsymbol{W}_{j,:}^{(i)}$ is the $j^{th}$ row of $\boldsymbol{W}^{(i)}$, $b_j^{(i)}$ is the $j^{th}$ component of $\boldsymbol{b}^{(i)}$. (Hinz & Van de Geer, 2019)*

For any $\boldsymbol{x}$, $h_i(\boldsymbol{x})$ can be rewritten as $h_i(\boldsymbol{x}) = \boldsymbol{W}^{(i)}(\boldsymbol{x})h_{i-1}(\boldsymbol{x}) + \boldsymbol{b}^{(i)}(\boldsymbol{x})$, where $\boldsymbol{W}^{(i)}(\boldsymbol{x})$ is a matrix with some rows of zeros and $\boldsymbol{b}^{(i)}(\boldsymbol{x})$ is a vector with some zeros. More precisely,

$$\boldsymbol{W}^{(i)}(\boldsymbol{x})_{j,:} = \begin{cases} \boldsymbol{W}^{(i)}_{j,:}, & \text{if } s_{h_i}(\boldsymbol{x})_j = 1; \\ \boldsymbol{0}, & \text{if } s_{h_i}(\boldsymbol{x})_j = 0. \end{cases} \qquad b^{(i)}(\boldsymbol{x})_j = \begin{cases} b^{(i)}_j, & \text{if } s_{h_i}(\boldsymbol{x})_j = 1; \\ 0, & \text{if } s_{h_i}(\boldsymbol{x})_j = 0. \end{cases} \qquad (4)$$

To conveniently represent activation patterns of multi-layer in a MLP, we denote $\boldsymbol{h}^{(i)} = \{h_1, ..., h_i\}$, where $h_i$ is defined in Eq.2, and $\boldsymbol{S}_{\boldsymbol{h}^{(i)}}(\boldsymbol{x}) = (\boldsymbol{s}_{h_1}(\boldsymbol{x}), ..., \boldsymbol{s}_{h_i}(\boldsymbol{x}))$, $\mathbb{S}(\boldsymbol{h}^{(i)}) = \{\boldsymbol{S}_{\boldsymbol{h}^{(i)}}(\boldsymbol{x})|\boldsymbol{x} \in \mathbb{R}^{n_0}\}$. Given a fixed $\boldsymbol{x}$, it is easy to prove that $h_i(\boldsymbol{x})$ is an affine transform ($i = 1, 2, \ldots, L$). Suppose that $\boldsymbol{s} \in \{0,1\}^{n_1} \times \cdots \times \{0,1\}^{n_{L-1}}$, $\boldsymbol{h} = \{h_1, ..., h_{L-1}\}$ and $\boldsymbol{h}(\boldsymbol{x})$ represents $h_{L-1}(\boldsymbol{x})$, if $\mathbb{D} = \{\boldsymbol{x}|\boldsymbol{S}_{\boldsymbol{h}}(\boldsymbol{x}) = \boldsymbol{s}\} \neq \emptyset$, then $f$ is an affine transform in $\mathbb{D}$. And it is easy to prove that there exists a linear region $\mathbb{D}'$ so that $\mathbb{D} \subseteq \mathbb{D}'$. Therefore we have $|\mathbb{P}(f)| \leq |\mathbb{S}(\boldsymbol{h})|$.

In our computational framework, histogram is a key concept and defined as follows.

**Definition 3.** *Define a histogram $\boldsymbol{v}$ as follows. (Hinz & Van de Geer, 2019)*

$$\boldsymbol{v} \in \mathbb{V} = \left\{ \boldsymbol{x} \in \mathbb{N}^{\mathrm{N}} | \, \| \boldsymbol{x} \|_1 = \sum_{j=0}^{\infty} x_j < \infty \right\} \qquad (5)$$

A histogram is used to represent a discrete distribution of $\mathbb{N}$. For example, the histogram of non-negative integers $\mathbb{G} = \{1, 0, 1, 4, 3, 2, 3, 1\}$ is $(1, 3, 1, 2, 1)^{\top}$. For convenience, let $v_i = \sum_{x \in \mathbb{G}} \mathbf{1}_{x=i}$ and the histogram of $G$ is denoted by $\mathrm{Hist}(\mathbb{G})$. We can then define an order relation.

**Definition 4.** *For any two histograms $\boldsymbol{v}, \boldsymbol{w}$, define the order relation $\preceq$ as follows. (Hinz & Van de Geer, 2019)*

$$\boldsymbol{v} \preceq \boldsymbol{w} :\Leftrightarrow \forall J \in \mathbb{N}, \sum_{j=J}^{\infty} v_j \leq \sum_{j=J}^{\infty} w_j \qquad (6)$$

It is obvious that any two histograms are not always comparable. But we can define a $\max$ operation such that $\boldsymbol{v}^{(i)} \preceq \max\left(\{\boldsymbol{v}^{(i)}|i \in I\}\right)$ where $I$ is an index collection. More precisely:

**Definition 5.** *For a finite index collection $I$, let $\mathbb{V}_I = \{\boldsymbol{v}^{(i)}|i \in I\}$, define $\max$ operation as follows.*

$$\max\left(\mathbb{V}_I\right)_J = \max_{i \in I} \left( \sum_{j=J}^{\infty} v_j^{(i)} \right) - \max_{i \in I} \left( \sum_{j=J+1}^{\infty} v_j^{(i)} \right) \quad \text{for } J \in \mathbb{N} \qquad (7)$$

*where $v_j^{(i)}$ is the $j^{th}$ component of the histogram $\boldsymbol{v}^{(i)}$. (Hinz & Van de Geer, 2019)*

When a region is divided by hyperplanes, the partitioned regions number will be affected by the space dimension which is defined as:

**Definition 6.** *For a connected and convex set $\mathbb{D} \subseteq \mathbb{R}^n$, if there exists a set of linear independent vectors $\left\{\boldsymbol{v}^{(i)}|\boldsymbol{v}^{(i)} \in \mathbb{R}^n, i = 1, \ldots, k, k \leq n\right\}$ and a fixed vector $\boldsymbol{c} \in \mathbb{R}^n$, s.t. (a) any $\boldsymbol{x} \neq \boldsymbol{c} \in \mathbb{D}$, $\boldsymbol{x} = \boldsymbol{c} + \sum_{i=1}^{k} a_i \boldsymbol{v}^{(i)}$ where $\{a_i\}$ are not all 0; (b) there exists $a_i \neq 0$ s.t. $a_i \boldsymbol{v}^{(i)} + \boldsymbol{c} \in \mathbb{D}$. Then the space dimension of $\mathbb{D}$ is $k$ and denote it as $\mathrm{Sd}(\mathbb{D}) = k$.*

The following proposition shows the change of space dimension after an affine transform.

**Proposition 1.** *Suppose $\mathbb{D} \subseteq \mathbb{R}^n$ is a connected and convex set with space dimension of $k$ ($k \leq n$), $f$ is an affine transform with domain of $\mathbb{D}$ and can be written as $f(\boldsymbol{x}) = \boldsymbol{A}\boldsymbol{x} + \boldsymbol{b}$, where $\boldsymbol{A} \in \mathbb{R}^{m \times n}$ and $\boldsymbol{b} \in \mathbb{R}^m$. Then $f(\mathbb{D})$ is a connected and convex set and*

$$\mathrm{Sd}(f(\mathbb{D})) \leq \min(k, \mathrm{rank}(\boldsymbol{A})) \qquad (8)$$

The proof of proposition 1 is given in Appendix A.1.1.

Now we can analyze the relationship between the change of space dimension and activation patterns. Let us first consider the 1st layer in an MLP $f$. $\boldsymbol{W}^{(1)}_{j,:}$ and $b^{(1)}_j$ construct a hyperplane ($\boldsymbol{W}^{(1)}_{j,:}\boldsymbol{x} +$

$b_j^{(1)} = 0$) in $\mathbb{R}^{n_0}$ and this hyperplane divides the whole region of $\mathbb{R}^{n_0}$ into two parts. One part corresponds to successful activation of the $j^{\text{th}}$ node in the 1st layer and the other to unsuccessful activation. This can be represented by the $j^{\text{th}}$ component of $\boldsymbol{s}_{h_1}(\boldsymbol{x})$. Therefore all the possible activation pattern $\boldsymbol{s}_{h_1}(\boldsymbol{x})$ is one by one correspondent to the divided regions by $n_1$ hyperplanes of $\{\boldsymbol{W}_{j,:}^{(1)}\boldsymbol{x}+b_j^{(1)} = 0, j = 1, \ldots, n_1\}$, which are denoted as $\mathbb{H}_{h_1}$. For any region $\mathbb{D}$ divided by $\mathbb{H}_{h_1}$, we have $h_1(\boldsymbol{x}) = \boldsymbol{W}^{(1)}(\boldsymbol{x}^{(0)})\boldsymbol{x}+\boldsymbol{b}^{(1)}(\boldsymbol{x}^{(0)})$ where $\boldsymbol{x}^{(0)}$ is any point in $\mathbb{D}$ and $\text{rank}(\boldsymbol{W}^{(1)}(\boldsymbol{x}^{(0)})) \leq |\boldsymbol{s}|_1$ where $\boldsymbol{s}$ is the correspondent activation pattern of $\mathbb{D}$. According to proposition 1, $h_1(\mathbb{D})$ satisfies that $\text{Sd}(h_1(\mathbb{D})) \leq \min\{n_0, |\boldsymbol{s}|_1\}$. Similarly, $\mathbb{H}_{h_2}$ divides $h_1(\mathbb{D})$ into different parts, and this corresponds to divide $\mathbb{D}$ into more sub-regions. In general, every element of $\mathbb{S}(\boldsymbol{h}^{(i)})$ is correspondent to one of regions that are partitioned by $\mathbb{H}_{h_1}, \mathbb{H}_{h_2}, \ldots, \mathbb{H}_{h_i}$. The next two definitions are used to describe the relationship between the change of space dimension and activation patterns.

**Definition 7.** *Define $\mathcal{H}_{\text{sd}}(\mathbb{S}_{\boldsymbol{h}})$ as the space dimension histogram of regions partitioned by a ReLU network defined by Eq.1 where $\boldsymbol{h} = \{h_1, \ldots, h_{L-1}\}$, i.e.*

$$\mathcal{H}_{\text{sd}}(\mathbb{S}_{\boldsymbol{h}}) = \text{Hist}\left(\{\text{Sd}\left(\boldsymbol{h}\left(\mathbb{D}\left(\boldsymbol{s}\right)\right)\right) \mid \mathbb{D}\left(\boldsymbol{s}\right) = \{\boldsymbol{x}|\boldsymbol{S}_{\boldsymbol{h}}\left(\boldsymbol{x}\right) = \boldsymbol{s}\}, \boldsymbol{s} \in \mathbb{S}\left(\boldsymbol{h}\right)\}\right). \tag{9}$$

**Definition 8.** *Define $\mathcal{H}_{\text{d}}\left(\mathbb{S}_{\boldsymbol{h}}\right)$ as the dimension histogram of regions partitioned by a ReLU network defined by Eq.1 where $\boldsymbol{h} = \{h_1, \ldots, h_{L-1}\}$, i.e.*

$$\mathcal{H}_{\text{d}}\left(\mathbb{S}_{\boldsymbol{h}}\right) = \text{Hist}\left(\{\min\left\{n_0, |\boldsymbol{s}_{h_1}|_1, ..., |\boldsymbol{s}_{h_{L-1}}|_1\right\} | \boldsymbol{s} \in \mathbb{S}(\boldsymbol{h})\}\right). \tag{10}$$

We then have the following proposition.

**Proposition 2.** *Given a ReLU network defined by Eq.1, let $\boldsymbol{h} = \{h_1, \ldots, h_{L-1}\}$, then $\mathcal{H}_{\text{sd}}(\mathbb{S}_{\boldsymbol{h}}) \preceq \mathcal{H}_{\text{d}}\left(\mathbb{S}_{\boldsymbol{h}}\right)$.*

The proof of Proposition 2 is given in the Appendix A.1.2. Proposition 2 shows that space dimension is limited by dimension histogram. This idea is used to compute the upper bound.

We can then start to introduce our computational framework. For convenience, we denote $\text{RL}(n, n')$ as one layer of a ReLU MLP with $n$ input nodes and $n'$ output nodes (containing one linear transform and one ReLU activation function), and define its activation histogram as follows.

**Definition 9.** *Given $h \in \text{RL}(n, n')$, define $\mathcal{H}_{\text{a}}(\mathbb{S}_h)$ as the activation histogram of regions partitioned by $h$, i.e.*

$$\mathcal{H}_{\text{a}}(\mathbb{S}_h) = \text{Hist}\left(\{|\boldsymbol{s}|_1 \mid \boldsymbol{s} \in \{\boldsymbol{s}_h\left(\boldsymbol{x}\right) \mid \boldsymbol{x} \in \mathbb{R}^n\}\}\right). \tag{11}$$

The activation histogram is similar to the dimension histogram, and it is used to define $\gamma$ which is a key parameter to construct an upper bound.

**Definition 10.** *If $\gamma'_{n,n}$ satisfies following conditions: (a) $\forall n' \in \mathbb{N}_+, n \in \{0, \ldots, n'\}$, $\max\left\{\mathcal{H}_{\text{a}}\left(\mathbb{S}_h\right)|h \in \text{RL}\left(n, n'\right)\right\} \preceq \gamma_{n,n'}$; (b) $\forall n' \in \mathbb{N}_+, n, \tilde{n} \in \{0, \ldots, n'\}, n \leq \tilde{n} \implies \gamma_{n,n'} \preceq \gamma_{\tilde{n},n'}$. Then, $\gamma_{n,n'}$ $(n' \in \mathbb{N}_+, n \in 1, ..., n')$ satisfies the bound condition. (Hinz & Van de Geer, 2019)*

Here, for $h \in \text{RL}(0, n')$, we define $\mathcal{H}(\mathbb{S}_h) = \boldsymbol{e}^0 = (1, 0, 0, \ldots)^\top$. Let $\Gamma$ be the set of all $(\gamma_{n,n'})_{n' \in \mathbb{N}_+, n \in \{0, \ldots, n'\}}$ that satisfy the bound conditions. When $n > n'$, $\gamma_{n,n'}$ is defined to be equal to $\gamma_{n',n'}$ since $\max\left\{\mathcal{H}_{\text{a}}\left(\mathbb{S}_h\right)|h \in \text{RL}\left(n, n'\right)\right\}$ is equal to $\max\left\{\mathcal{H}_{\text{a}}\left(\mathbb{S}_h\right)|h \in \text{RL}\left(n', n'\right)\right\}$ (Hinz & Van de Geer, 2019). By the definition $\gamma_{n,n'}$ represents an upper bound of the activation histogram of regions which are derived from $n$-dimension space partitioned by $n'$ hyperplanes. According to Proposition 1, this upper bound is also related to the upper bound of space dimension. Therefore when $\gamma_{n,n'}$ is tighter the computation of upper bound of linear regions number will be more accurate. The following function is used to describe the relationship between upper bounds of activation histogram and space dimension.

**Definition 11.** *For $i^* \in \mathbb{N}$, define a clipping function $\text{cl}_{i^*}(\cdot) : \mathbb{V} \to \mathbb{V}$ as follows. (Hinz & Van de Geer, 2019)*

$$\text{cl}_{i^*}(\boldsymbol{v})_i = \begin{cases} v_i & \text{for } i < i^* \\ \sum_{j=i^*}^{\infty} v_j & \text{for } i = i^* \\ 0 & \text{for } i > i^* \end{cases} \tag{12}$$

With the definitions and notations above, we can introduce the computational framework to compute the upper bound of linear regions number as follows.

**Proposition 3.** *For a $\gamma \in \Gamma$, define the matrix $\boldsymbol{B}_{n'}^{(\gamma)} \in \mathbb{N}^{(n'+1)\times(n'+1)}$ as*

$$\left(B_{n'}^{(\gamma)}\right)_{i,j} = (\mathrm{cl}_{j-1}\left(\gamma_{j-1,n'}\right))_{i-1}, \quad i,j \in \{1,\ldots,n'+1\}.$$

*Then, the upper bound of linear regions number of an MLP in Eq.1 is*

$$\left\| \boldsymbol{B}_{n_{L-1}}^{(\gamma)} \boldsymbol{M}_{n_{L-2},n_{L-1}} \ldots \boldsymbol{B}_{n_1}^{(\gamma)} \boldsymbol{M}_{n_0,n_1} \boldsymbol{e}^{n_0+1} \right\|_1, \tag{13}$$

*where $\boldsymbol{M}_{n,n'} \in \mathbb{R}^{(n'+1)\times(n+1)}, (M)_{i,j} = \delta_{i,\min(j,n'+1)}$. (Hinz & Van de Geer, 2019)*

## 3 MAIN RESULTS

In this section, we introduce a better choice of $\gamma$ and compare it to Serra et al. (2018). We also extend the computational framework to some widely used network structures.

### 3.1 A TIGHTER UPPER BOUND ON THE NUMBER OF REGIONS

Before giving our main results, we define a new function of histograms as follows.

**Definition 12.** *Define a downward-move function $\mathrm{dm}(\cdot) : \mathbb{V} \to \mathbb{V}$ by $\mathrm{dm}(\boldsymbol{v})_i = v_{i-1}$. When $i = 0$, set $\mathrm{dm}(\boldsymbol{v})_0 = 0$.*

We then prove the following two theorems used to compute $\gamma$.

**Theorem 1.** *If $\gamma_{n,n''}$ satisfies the bound condition in Definition 10 when $n'' < n'$, then*

$$\gamma_{n,n'} = \gamma_{n-1,n'-1} + \mathrm{dm}(\gamma_{n,n'-1}) \tag{14}$$

*also satisfies the bound condition.*

**Theorem 2.** *Given any $h \in \mathrm{RL}(1,n)$, its activation histogram $\mathcal{H}_{\mathrm{a}}(\mathbb{S}_h)$ denoted by $\boldsymbol{v}$ satisfies $\sum_{i=t}^{n} v_i \leq 2(n-t) + 1$.*

The proofs of Theorem 1 and 2 are in Appendix A.1.3 and A.1.4 respectively.

From Theorem 2, we can derive a feasible choice of $\gamma_{1,n}$ whose form is

$$\gamma_{1,n} = (\underbrace{0,\ldots,0}_{\lceil \frac{n}{2} \rceil - 1}, n \bmod 2, \underbrace{2,\ldots,2}_{\lfloor \frac{n}{2} \rfloor}, 1)^\top \tag{15}$$

For example, $\gamma_{1,4} = (0,0,2,2,1)^\top$. Since there exists $h \in \mathrm{RL}(1,n)$ such that $\mathcal{H}_{\mathrm{a}}(\mathbb{S}_h) = \gamma_{1,n}$ (the proof can be seen in the Appendix A.1.5), Eq.15 is the tightest upper bound, i.e. $\max\{\mathcal{H}_{\mathrm{a}}(\mathbb{S}_h) | h \in \mathrm{RL}(1,n)\}$ satisfies Eq.15. With the fact that $\max\{\mathcal{H}_{\mathrm{a}}(\mathbb{S}_h) | h \in \mathrm{RL}(2,1)\} = (1,1)^\top$, any $\gamma_{n,n'}$ can be computed.

### 3.2 COMPARISON WITH OTHER BOUNDS

We use $\gamma_{n,n'}^{\mathrm{ours}}$ and $\gamma_{n,n'}^{\mathrm{serra}}$ to represent the ones proposed by us and Serra et al. (2018). According to Hinz & Van de Geer (2019),

$$(\gamma_{n,n'}^{\mathrm{serra}})_i = \begin{cases} 0, & 0 \leq i < n' - n \\ \binom{n'}{i}, & n' - n \leq i \leq n' \end{cases}$$

It is easy to verify that $\gamma_{n,n'}^{\mathrm{serra}}$ also satisfies Eq.14. But the initial value of $\gamma_{n,n'}^{\mathrm{serra}}$ is different from $\gamma_{n,n'}^{\mathrm{ours}}$ and we have that $\gamma_{1,n}^{\mathrm{serra}} = (0,...,0,n,1)^\top$. Obviously, $\gamma_{1,n}^{\mathrm{ours}} \preceq \gamma_{1,n}^{\mathrm{serra}}$. Since the two operations, $+$ and $\mathrm{dm}(\cdot)$, keep the relation of $\preceq$, for any $n, n' \in \mathbb{N}_+$, $\gamma_{n,n'}^{\mathrm{ours}} \preceq \gamma_{n,n'}^{\mathrm{serra}}$ (see an example in Appendix A.2.1). By the following theorem, the upper bound computed by Eq.1 using $\gamma_{n,n'}^{\mathrm{ours}}$ is tighter than $\gamma_{n,n'}^{\mathrm{serra}}$.

**Theorem 3.** *Given $\gamma^{(1)}, \gamma^{(2)} \in \Gamma$, if for any $n, n' \in \mathbb{N}_+$ such that $\gamma_{n,n'}^{(1)} \preceq \gamma_{n,n'}^{(2)}$, then upper bound computed by Eq.13 using $\gamma^{(1)}$ is less than or equal to the one using $\gamma^{(2)}$.*

The proof of Theorem 3 is in Appendix A.1.6. Theorem 3 shows that if we can get a "smaller" $\gamma_{n,n'}$ then the upper bound will be tighter. And Theorem 1 implies that if we have a more accurate initial value, e.g. "smaller" $\gamma_{2,n}$ or $\gamma_{3,n}$, the upper bound could be further improved. Therefore Theorem 1 provides a potential approach to achieve even tighter bound.

### 3.3 EXPANSION TO COMMON NETWORK STRUCTURES

Proposition 3 is only applied for ReLU MLPs. In this section we extend it to widely used network structures by introducing the corresponding matrix computation.

**Pooling an unpooling layers**    Since the pooling layer or unpooling layer can be written as a linear transform., e.g. average-pooling and linear interpolation, it can be denoted by $\boldsymbol{y} = A\boldsymbol{x}$. Suppose $\boldsymbol{v}$ is the space dimension of input region and $\boldsymbol{w}$ is the histogram of the output regions. Then we have the following proposition.

**Proposition 4.** *Suppose* $\mathrm{rank}(\boldsymbol{A}) = k$, *then* $\boldsymbol{w} \preceq \mathrm{cl}_k(\boldsymbol{v})$.

Since $\mathrm{cl}_k(\boldsymbol{v})$ has a matrix computation form which is similar to $M$ in Eq.13, the effect of this type of layer on the space dimension histogram can be regarded as another matrix in Eq.13. Similarly, we have the following proposition for Max-pooling layers.

**Proposition 5.** *Suppose the max-pooling layer is equivalent to a* $k$-rank *maxout layer with* $n$ *input nodes and* $n_l$ *output nodes. Let* $c = (k^2 - k)n_l$, *then*

$$\boldsymbol{w} \preceq \mathrm{cl}_{n_l} \left( \mathrm{diag} \left\{ |\gamma_{0,c}|_1, \ldots, |\gamma_{n,c}|_1 \right\} \boldsymbol{v} \right) \tag{16}$$

The proofs of Proposition 4, Proposition 5 are in Appendix A.1.7 and A.1.8.

**Skip connection and residual structure**    The skip connection is very popular in current network architecture design. When a network is equipped with a skip connection, the upper bound will be changed. The following Proposition gives the correspondent method to compute the upper bound with skip connection.

**Proposition 6.** *Given a network, suppose that* $\boldsymbol{v}$ *is the space dimension histogram of the input regions of the* $i^{th}$ *layer and* $\boldsymbol{w}$ *the histogram of the output regions of the* $j^{th}$ *layer. And we have that* $\boldsymbol{w} \preceq \prod_{k=i}^{j} \boldsymbol{A}_k \boldsymbol{v}$. *When the input of the* $i^{th}$ *layer is concatenated to the* $j^{th}$ *layer, i.e. a skip connection, then the computation of* $\boldsymbol{w}$ *will change and satisfies that* $\boldsymbol{w} \preceq \sum v_n |\boldsymbol{B}\boldsymbol{e}^n|_1 \boldsymbol{e}^n$ *where* $\boldsymbol{B} = \prod_{k=i}^{j} \boldsymbol{A}_k$. *It also has a matrix multiplication form.*

Residual structures (He et al., 2016) are similar to skip connections and can be regarded as a skip connection plus a simple full-rank square matrix computation. Therefore its effect on space dimension histogram is the same as the skip connection. Concatenation is equivalent to addition in upper bound computation. Thus, we have the following proposition.

**Proposition 7.** *Suppose that the residual structure adds the input of the* $i^{th}$ *layer to the output of the* $j^{th}$ *layer.* $\boldsymbol{v}, \boldsymbol{w}, \boldsymbol{B}$ *have the same meaning in Proposition 6. Then we have that* $\boldsymbol{w} \preceq \sum v_n |\boldsymbol{B}\boldsymbol{e}^n|_1 \boldsymbol{e}^n$.

The proofs of Proposition 6 and Proposition 7 are in Appendix A.1.9 and A.1.10. In addition, the analysis for dense connections (Huang et al., 2017) is similar to skip connections and is further discussed in Section 5.

With the above analysis, we can deal with more complex network. For example, a U-net (Ronneberger et al., 2015) is composed of convolutional layers, pooling layers, unpooling layers and skip connections. If we regard convolutional layers as fully-connected layers, then we could compute the upper bound of it (see details in Appendix A.2.3). Though in this paper we have not extend to all possible network structures and architecture, according to Proposition 1 and by using our computational framework, it is possible to compute the change of the space dimension histogram for any other network structure and architecture. Based on these bricks of different layers or structures, we may derive the upper bound of much more complex and practically used networks.

## 4    EXPERIMENTS

We perform two experiments in this work. The first one is to compare the bound proposed by us and by Serra et al. (2018) which is illustrated by Figure 1. The upper bounds of MLPs computed and their ratio is calculated to measure the difference. Figure1(a) and Figure 1(b) show how the ratio

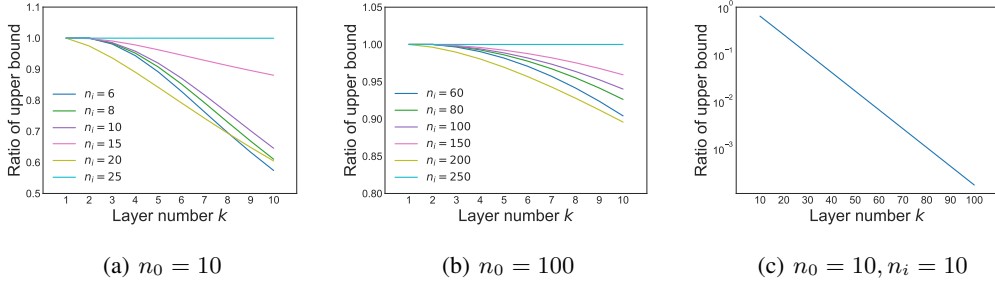

(a) $n_0 = 10$                 (b) $n_0 = 100$                 (c) $n_0 = 10, n_i = 10$

Figure 1: The comparison of our upper bound and Serra et al. (2018). The $y$ axis represents the ratio of them which can be used to measure the difference. The MPLs are in the form of $n_0$-$n_i$-$n_i$-$\cdots$-$n_i$-1 with $k$ hidden layers. (a) Setting is $n_0 = 10, n_i = 6, 8, 10, 15, 20, 25, k = 1, 2, ..., 10$. (b) Setting is $n_0 = 10, n_i = 6, 8, 10, 15, 20, 25, k = 1, 2, ..., 10$. (c) The setting is $n_0 = n_i = 10, k = 10, 20, ..., 100$.

Table 1: The upper bound of AEs and U-nets. The channels setting means the architecture of networks. For instance, 4-8-16-32 represents an AE which encoder has three down-sampling layers and the channels in different layers are 4, 8, 16, 32 respectively (see details in Appendix A.3.1). The input size in the first four experiments is $(24 \times 24)$ while the last two is $(16 \times 16)$. The upper bounds are listed in the third and fourth column. The former corresponds to U-nets while the latter to AEs. Besides, the ratios of two them are listed in the last column.

| No. | Channels setting | w/ skip connection | w/o skip connection | ratio |
|---|---|---|---|---|
| 1 | 4-8-16-32 | $5.431 \times 10^{2031}$ | $5.027 \times 10^{1626}$ | $1.080 \times 10^{405}$ |
| 2 | 4-16-64-128 | $1.712 \times 10^{3111}$ | $3.333 \times 10^{2956}$ | $3.517 \times 10^{154}$ |
| 3 | 4-8-16 | $9.354 \times 10^{1901}$ | $1.407 \times 10^{1643}$ | $6.648 \times 10^{258}$ |
| 4 | 4-16-64 | $2.211 \times 10^{2591}$ | $3.328 \times 10^{2450}$ | $6.644 \times 10^{140}$ |
| 5 | 8-16-64 | $3.504 \times 10^{1395}$ | $5.470 \times 10^{1330}$ | $6.406 \times 10^{64}$ |
| 6 | 8-32-128 | $1.268 \times 10^{1642}$ | $1.230 \times 10^{1642}$ | $1.031$ |

changes as the number of hidden layers increases from 1 to 10 with different $n_0$. Figure 1(c) shows how the ratio changes when the depth of the network becomes larger.

The second one is to verify the effectiveness of skip connections and residual structures on the improvement of expressiveness, as illustrated in Table 1 and Table 2. We compute the upper bound of auto-encoders (AE) and U-nets. AEs and correspondent U-nets have the same network architecture except the skip connection. As for residual structures, we build two identical networks for the image classification task with or without residual structure. For each pair of networks (with or without one special structure), the ratio of their bounds are computed to measure how the upper bound will be enhanced with special structure. Different network architecture settings are tried in experiments. Because of the large memory required by complete $\gamma_{n,n'}$, we only consider networks of relatively small sizes (network input size is $24 \times 24$ or $16 \times 16$). In addition, convolutional layers are regarded as fully-connected layers in the computation.

## 5 DISCUSSION

The proposed upper bound has been theoretically proved to be tighter than Serra et al. (2018). Furthermore, Figure 1(a) and Figure 1(b) show that when $n_i$ increases from $0.6n_0$ to $1.5n_0$, the ratio curve of two bounds moves upward. However, when $n_i$ increases to $2n_0$, the curve moves downward. And when $n_i$ increases further and is much larger than $n_0$, the two bounds are the same. This trend is related to the extent of similarity in $\boldsymbol{B}_n^{(\text{ours})}$ and $\boldsymbol{B}_n^{(\text{serra})}$. When $n_i < n_0$, $\boldsymbol{B}_{n_i}$ affects the upper bound most. And the difference between $\boldsymbol{B}_{n_i}^{(\text{ours})}$ and $\boldsymbol{B}_{n_i}^{(\text{serra})}$ is more significant when $n_i$ decreases. When $n_i > n_0$, $(\boldsymbol{B}_n)_{:,n_0+1}$, the $(n_0 + 1)^{\text{th}}$ column of $B_n$, is the main source of

Table 2: The upper bound of a simple network for the classification task with or without residual structure. The channels setting represents where main differences in architectures are (see details in Appendix A.3.2). "p16" means that there is a pooling layer before the convolutional layer with 16 channels. And "r16" represents that a residual structure is added in the convolutional layer with 16 channels. The last three columns in this table is similar to Table 1. In this part, the input size in all the experiments is $(24 \times 24)$.

| No. | Channels setting | w/ res-structure | w/o res-structure | ratio |
|---|---|---|---|---|
| 1 | 4-p16-p16-r16-r16-r16 | $7.313 \times 10^{1930}$ | $3.720 \times 10^{1930}$ | 1.966 |
| 2 | 4-p4-4-r4-r4-r4 | $5.052 \times 10^{1542}$ | $3.393 \times 10^{1542}$ | 1.489 |
| 3 | 4-p16-p16-16-16-16 | $5.232 \times 10^{1930}$ | $3.720 \times 10^{1930}$ | 1.406 |
| 4 | 4-p16-p16-r16-r16-r16-r16-r16 | $3.012 \times 10^{2277}$ | $5.882 \times 10^{2276}$ | 5.121 |
| 5 | 4-p16-p32-r32-r32-r32 | $7.180 \times 10^{2623}$ | $7.166 \times 10^{2623}$ | 1.002 |

difference. It is easy to find that the middle part of $B_n^{(\text{ours})}$ and $B_n^{(\text{serra})}$ differ most while the left and right part are similar (see detials in Appendix A.2.3). As for the Figure 1(c), it shows the difference between two upper bounds will be enlarged when the depth of networks increases. In general, both theoretical analysis and experimental results show our bound is tighter than Serra et al. (2018).

The result in Table 1, Table 2 shows that special structure such as skip connection and residual structure will enhance the upper bound. And when the residual structure is used in more layers, the enhancement will increase (comparing comparing No.1, 3 and 4 in Table 2). However, it seems that when the number of channels is larger, the enhancement is weaker (comparing No.1 to 2, 3 to 4, and No.5 to No.6 in Tabel 1). Similar result can be observed in Table 2 (comparing No.1 to No.5). These results explain why skip connection and residual structure are effective from the perspective of expressiveness. However, there also exist some settings such that the two upper bounds are close.

We provide an explanation for the observation through the concept of partition efficiency. First of all, we note that it is not the upper bound but the practical number of linear regions that is directly related to the network expressiveness. Though upper bound measures the expressiveness in some extent, there is a gap between upper bound and the real practical maximum. Actually, any upper bound which can be computed by the framework of matrix computation, including Pascanu et al. (2013); Montufar et al. (2014); Serra et al. (2018) etc., assume that all the output regions of the current layer will be further divided at the same time by all the hyperplanes which the next layer implies. However, this assumption is not always sound (see detail in Appendix A.2.4). When the number of regions far exceeds the dimension of hyperplanes, it is hard to imagine that one hyperplane can partition all the regions. But the special structure (e.g. skip connection) may increase the number of regions partitioned by one hyperlane such that the practical number can be increased even if the upper bound is not enhanced. Intuitively, skip connections increase the dimension of input space and the partition efficiency may be higher. An good example is dense connections (Huang et al., 2017) which is only a special case of skip connection but can further increase the dimension and thus lead to better expressiveness; this may explain the success of DenseNets. Though we have not completely proved the partition efficiency of skip connections, some evidences seem to confirm our intuition, as shown in Appendix A.1.11 and A.1.12.

## 6 CONCLUSION

In this paper, we study the expressiveness of ReLU DNNs using the upper bound of number of regions that are partitioned by the network in input space. We then provide a tighter upper bound than previous work and propose a computational framework to compute the upper bounds of more complex and practically more useful network structures. Our experiments verify that special network structures (e.g. skip connection and residue structure) can enhance the upper bound. Our work reveals that the number of linear regions is a good measurement of expressiveness, and it may guide us to design more efficient new network architectures since our computational framework is able to practically compute its upper bound.

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

## A  APPENDIX

### A.1  PROOFS

#### A.1.1  THE PROOF OF PROPOSITION 1

*Proof.* By Definition 6, suppose $\{v^{(1)}, v^{(2)}, \ldots, v^{(k)}\}$ and $c$ can be used to represent any $x \in \mathbb{D}$ and $\{a_i \neq 0\}$ satisfies that $a_i\{v^{(i)} + c \in \mathbb{D}\}$. Let $\mathbb{D}' = \mathbb{D} - c$, then $\mathbb{D}'$ is a translation of $\mathbb{D}$. Since $\mathbb{D}$ is convex, $\mathbb{D}'$ and $f(\mathbb{D}')$ is also convex. Because $f(\mathbb{D}) = f(\mathbb{D}' + c) = A\mathbb{D}' + Ac + b$, then $\mathrm{Sd}(f(\mathbb{D})) = \mathrm{Sd}(A\mathbb{D}')$. Suppose that $\mathbb{D}'' = A(\mathbb{D}') = \{Ax | x \in R'\}$. For any $y \in \mathbb{D}''$, there exists $x \in \mathbb{D}'$, s.t. $y = Ax$. Denote $Av^{(i)}$ by $w^{(i)}$, then $y$ can be represented by $\{w^{(1)}, w^{(2)}, \ldots, w^{(k)}\}$ and $a_i w^{(i)} \in \mathbb{D}''$. From $\{w^{(1)}, w^{(2)}, \ldots, w^{(k)}\}$ choose a set of vectors which are linear independent and can linearly represent $\{w^{(1)}, w^{(2)}, \ldots, w^{(k)}\}$. For convenience, suppose they are $\{w^{(1)}, w^{(2)}, \ldots, w^{(t)}\}$, $t \leq k$. Therefore any $y \in \mathbb{D}''$ can be represented by $\{w^{(1)}, w^{(2)}, \ldots, w^{(t)}\}$. Thus, $\mathrm{Sd}(\mathbb{D}'') = \mathrm{Sd}(f(\mathbb{D})) = t$ and we have that

$$
\begin{aligned}
t &= \mathrm{rank}(\begin{bmatrix} w^{(1)} & \ldots & w^{(t)} \end{bmatrix}) \\
&= \mathrm{rank}(\begin{bmatrix} w^{(1)} & \ldots & w^{(k)} \end{bmatrix}) \\
&= \mathrm{rank}(A \begin{bmatrix} v^{(1)} & \ldots & v^{(k)} \end{bmatrix}) \\
&\leq \min\{\mathrm{rank}(A), \mathrm{rank}(\begin{bmatrix} v^{(1)} & \ldots & v^{(k)} \end{bmatrix})\} \\
&= \min\{\mathrm{rank}(A), k\}.
\end{aligned}
$$

The last equality is derived from that $\{v^{(1)}, v^{(2)}, \ldots, v^{(k)}\}$ are $k$ linear independent vectors. $\square$

#### A.1.2  THE PROOF OF PROPOSITION 2

*Proof.* For any $s \in \mathbb{S}(h)$, suppose $\mathbb{D}(s) = \{x | S_h(x) = s\}$. As long as $\mathrm{Sd}(h(\mathbb{D}(s))) \leq \min\{n_0, |s_{h_1}|_1, |s_{h_2}|_1, \ldots, |s_{h_{L-1}}|_1\}$ is established, the proposition can be proved. Apparently, $\mathrm{Sd}(\mathbb{D}(s)) = n_0$. Through the analysis in Section 2, $h(\cdot)$ in $\mathbb{D}(s)$ is an affine transform. For any $x^{(1)}, x^{(2)} \in \mathbb{D}(s)$, $W^{(i)}(x^{(1)}) = W^{(i)}(x^{(2)}), b^{(i)}(x^{(1)}) = b^{(i)}(x^{(2)})$. Therefore we use $x^{(0)} \in \mathbb{D}(s)$ to represent them. Then $h(\cdot)$ in $\mathbb{D}(s)$ can be written as

$$
\begin{aligned}
h(x) &= W^{(L-1)}(x^{(0)})(\ldots W^{(1)}(x^{(0)})x + b^{(1)}(x^{(0)})) + b^{(L-1)}(x^{(0)}) \\
&= W(x^{(0)})x + b(x^{(0)})
\end{aligned}
$$

By Proposition 1, we have

$$\mathrm{Sd}\left(\boldsymbol{h}\left(\mathbb{D}\left(\boldsymbol{s}\right)\right)\right) \leq \min\left\{\mathrm{rank}(\boldsymbol{W}(\boldsymbol{x}^{(0)})), n_0\right\}$$

$$= \min\left\{n_0, \mathrm{rank}\left(\prod_{i=1}^{L-1} \boldsymbol{W}^{(i)}(\boldsymbol{x}^{(0)})\right)\right\}$$

$$\leq \min\left\{n_0, \min\left\{\mathrm{rank}\left(\boldsymbol{W}^{(1)}(\boldsymbol{x}^{(0)})\right), \ldots, \mathrm{rank}\left(\boldsymbol{W}^{(L-1)}(\boldsymbol{x}^{(0)})\right)\right\}\right\}$$

$$\leq \min\{n_0, |\boldsymbol{s}_{h_1}|_1, |\boldsymbol{s}_{h_2}|_1, ..., |\boldsymbol{s}_{h_{L-1}}|_1\}$$

The last two inequality are derived from that

$$\mathrm{rank}\left(\prod_{i=1}^{L-1} \boldsymbol{W}^{(i)}(\boldsymbol{x}^{(0)})\right) \leq \min\{\mathrm{rank}(\boldsymbol{W}^{(1)}(\boldsymbol{x}^{(0)})), ..., \boldsymbol{W}^{(L-1)}(\boldsymbol{x}^{(0)}))\}$$

and $\mathrm{rank}(\boldsymbol{W}^{(i)}(\boldsymbol{x}^{(0)})) \leq |\boldsymbol{s}_{h_i}|_1$. □

### A.1.3 THE PROOF OF THEOREM 1

*Proof.* Obviously, $\gamma_{n,n'}$ satisfies the second condition. As for the first condition, consider that hyperplanes $\{L_1, L_2, ..., L_{n'}\}$ divide $\mathbb{R}^n$. These hyperlanes correspond to one $h \in \mathrm{RL}(n, n')$ and its activation histogram is denoted by $\boldsymbol{v}_1$. Here, we assume that $L_1, L_2, \ldots, L_{n'}$ are not parallel to each other since in parallel case it is easy to imagine and verify that there exists a $h' \in \mathrm{RL}(n, n')$ which satisfies $\mathcal{H}_a(\mathbb{S}_h) \preceq \mathcal{H}_a(\mathbb{S}_{h'})$. Suppose that hyperplanes $\{L_1, L_2, \ldots, L_{n'-1}\}$ divide $\mathbb{R}^n$ into $t$ regions $\{R_1, R_2, ..., R_t\}$ and their activation histogram is denoted by $\boldsymbol{v}_2$. Assume that $L_{n'}$ crosses $\{R_1, R_2, ..., R_p\}, p \leq t$ and divides them into $2p$ regions. Let $\boldsymbol{v}_3$ denote the activation histogram of $\{R_1, R_2, \ldots, R_p\}$. $p$ regions of them are active by $L_{n'}$ while other $p$ regions are not. Part of $\{R_{p+1}, R_{p+2}, ..., R_t\}$ are active by $L_{n'}$ while the rest regions are not. Their activation histogram are denoted as $\boldsymbol{v}_4$ and $\boldsymbol{v}_5$ respectively. Then we have the following equations:

$$\boldsymbol{v}_2 = \boldsymbol{v}_3 + \boldsymbol{v}_4 + \boldsymbol{v}_5 \tag{17}$$

$$\boldsymbol{v}_1 = \boldsymbol{v}_3 + \mathrm{dm}(\boldsymbol{v}_3) + \mathrm{dm}(\boldsymbol{v}_4) + \boldsymbol{v}_5 \tag{18}$$

Let us only focus on $\{R_1.R_2, ..., R_p\}$. Suppose $\{L_1, L_2, ..., L_m\}(m \leq n'-1)$ are borders of these regions. $\{L_{n'} \cap L_1, L_{n'} \cap L_2, ..., L_{n'} \cap L_m\}$ are hyperplanes in $L_{n'}$, and their active directions are projections of $\{L_1, L_2, ..., L_m\}$ in $\mathbb{R}^n$. So the activation histogram of $\{L_{n'} \cap L_1, L_{n'} \cap L_2, ..., L_{n'} \cap L_m\}$ in $L_{n'}$ which is denoted by $\boldsymbol{v}_6$ is equal to $\boldsymbol{v}_3$. By the assumption that when $n'' < n'$, $\gamma_{n,n''}$ satisfies the bound condition, we have

$$\boldsymbol{v}_3 = \boldsymbol{v}_6 \preceq \gamma_{n-1,m} \preceq \gamma_{n-1,n'-1}. \tag{19}$$

By Eq.17, Eq.18 and Eq.19, $\boldsymbol{v}_1$ satisfies

$$\boldsymbol{v}_1 \preceq \gamma_{n-1,n'-1} + \mathrm{dm}(\boldsymbol{v}_3 + \boldsymbol{v}_4 + \boldsymbol{v}_5)$$
$$= \gamma_{n-1,n'-1} + \mathrm{dm}(\boldsymbol{v}_2) \tag{20}$$
$$\preceq \gamma_{n-1,n'-1} + \mathrm{dm}(\gamma_{n,n'-1})$$

Because of the arbitrariness of $\boldsymbol{v}_1$, we can get $\gamma_{n,n'} = \gamma_{n-1,n'-1} + \mathrm{dm}(\gamma_{n,n'-1})$ which satisfies the first condition. □

### A.1.4 THE PROOF OF THEOREM 2

*Proof.* For 1-dimension space, its hyperlane is one point in the number axis and the activation direction is either left or right. It is apparent that $n$ segmentation points divide one line into $n+1$ parts. And for any $x \in \mathbb{R}$, its activation number is at most $n$. Therefore if $t < \lceil \frac{n}{2} \rceil$, then $\sum_{i=t}^n v_i \leq n+1 \leq 2(n-t)+1$. If $t \geq \lceil \frac{n}{2} \rceil$, the activation pattern of $n+1$ regions in $\mathbb{R}$ is denoted by $\{\boldsymbol{s}_1, \boldsymbol{s}_2, \ldots, \boldsymbol{s}_{n+1}\}$ (see Figure 2(a)). No matter how the activation directions of $n$ points are, we have $|\boldsymbol{s}_1| + |\boldsymbol{s}_{n+1}| = n$ since $|\boldsymbol{s}_1|$ is equal to the number of left activation directions and $|\boldsymbol{s}_{n+1}|$ is equal to the number of right activation direction. Another obvious conclusion is that

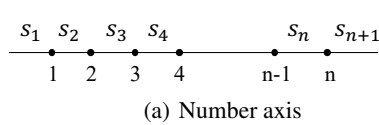

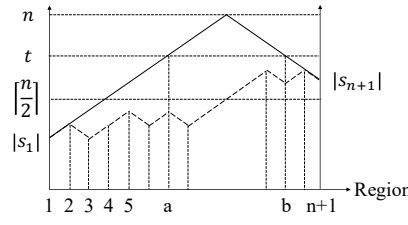

(a) Number axis

(b) Activation number of each region

Figure 2: (a) The number axis is partitioned into $n + 1$ parts. $s_i$ represents the $i^{\text{th}}$ region. (b). An example of a partition and the activation number of each region. The abscissa axis corresponds to the position and the vertical axis represents the number of activation

$|s_i| - |s_{i+1}| = \pm 1$ as the activation patterns of neighboring regions only differ at the $i^{\text{th}}$ point. Without loss of generality, let $|s_1| \leq \lceil \frac{n}{2} \rceil$. Any line chart of active numbers with dotted line is under the line chart with solid line (see Figure 2(b)). So regions with active numbers larger than or equal to $t$ are among $a^{th}$ and $b^{th}$ region. Apparently, the number of regions among $a^{th}$ and $b^{th}$ region is $2(n - t) + 1$. So $\sum_{i=t}^{n} v_i \leq 2(n - t) + 1$. $\qquad \square$

### A.1.5  THE PROOF OF EQ.15

*Proof.* Suppose the $n$ points of $h$ in the number axis are $p_1, p_2, ..., p_n$. Let the direction sof $p_1, ..., p_{\lfloor \frac{n}{2} \rfloor}$ are right and other points left. It is easy to verify that $\mathcal{H}_a (\mathbb{S}_h) = \gamma_{1,n}$ where $\gamma_{1,n}$ is defined by

$$\gamma_{1,n} = (\underbrace{0, \ldots, 0}_{\lceil \frac{n}{2} \rceil - 1}, n \bmod 2, \underbrace{2, \ldots, 2}_{\lfloor \frac{n}{2} \rfloor}, 1)^{\top}$$

Then $\gamma_{1,n} \preceq \max \{ \mathcal{H}_a (\mathbb{S}_h) \, | h \in \mathrm{RL}(1, n) \}$. Since $\max \{ \mathcal{H}_a (\mathbb{S}_h) \, | h \in \mathrm{RL}(1, n) \} \preceq \gamma_{1,n}$, $\gamma_{1,n} = \max \{ \mathcal{H}_a (\mathbb{S}_h) \, | h \in \mathrm{RL}(1, n) \}$. $\qquad \square$

### A.1.6  THE PROOF OF THEOREM 3

*Proof.* Different $\boldsymbol{B}_{n'}^{\gamma}$ can be derived from different $\gamma_{n,n'}$. Since the clipping function keep the order relation $\preceq$, every column of $\boldsymbol{B}_{n'}^{\gamma^{(1)}}$ and $B_{n'}^{\gamma^{(2)}}$ in the same position satisfy that

$$\left( \boldsymbol{B}_{n'}^{\gamma^{(1)}} \right)_{:,j} \preceq \left( \boldsymbol{B}_{n'}^{\gamma^{(2)}} \right)_{:,j}$$

Another fact is that when $\boldsymbol{v} \preceq \boldsymbol{w}$, then $|\boldsymbol{v}|_1 \leq |\boldsymbol{w}|_1$. Here we prove that if $\boldsymbol{v} \preceq \boldsymbol{w}$, then

$$\boldsymbol{M}_{n_i, n_{i+1}} \boldsymbol{v} \preceq \boldsymbol{M}_{n_i, n_{i+1}} \boldsymbol{w} \tag{21}$$

and

$$\boldsymbol{B}_{n_i}^{\gamma^{(1)}} \boldsymbol{v} \preceq \boldsymbol{B}_{n_i}^{\gamma^{(2)}} \boldsymbol{w} \tag{22}$$

Then the theorem can derived easily from Eq.21 and Eq.22.

Because $\boldsymbol{M}_{n_i, n_{i+1}} \boldsymbol{v} = \mathrm{cl}_{n_{i+1}}(\boldsymbol{v})$, Eq.21 is easy to verified. For $\boldsymbol{B}_{n_i}^{\gamma^{(k)}}, k = 1, 2$, we have that $\left( \boldsymbol{B}_{n_i}^{\gamma^{(k)}} \right)_{:,j_1} \preceq \left( \boldsymbol{B}_{n_i}^{\gamma^{(k)}} \right)_{:,j_2}$ when $j_1 \leq j_2$ because of the second condition in Definition 10 and the property of the clipping function. For convenience, $\left( B_{n_i}^{\gamma^{(k)}} \right)_{m,j}$ is denoted by $B_{m,j}^{(k)}, k = 1, 2$. By

Definition 5, we have that

$$\sum_{m=M}^{\infty} \left(\boldsymbol{B}^{(1)}\boldsymbol{v}\right)_m \leq \sum_{m=M}^{\infty} \left(\boldsymbol{B}^{(2)}\boldsymbol{w}\right)_m$$

$$\iff \sum_{m=M}^{\infty} \left(\sum_{j=0}^{\infty} B_{m,j}^{(1)} v_j\right)_m \leq \sum_{m=M}^{\infty} \left(\sum_{j=0}^{\infty} B_{m,j}^{(2)} w_j\right)_m$$

$$\iff \sum_{j=0}^{\infty} \left(\sum_{m=M}^{\infty} B_{m,j}^{(1)}\right) v_j \leq \sum_{j=0}^{\infty} \left(\sum_{m=M}^{\infty} B_{m,j}^{(2)}\right) w_j$$

Because $\boldsymbol{B}_{:,j}^{(1)} \preceq \boldsymbol{B}_{:,j}^{(2)}$, i.e. $\forall M \geq 0, \sum_{m=M}^{\infty} B_{m,j}^{(1)} \leq \sum_{m=M}^{\infty} B_{m,j}^{(2)}$. Let $a_j = \sum_{m=M}^{\infty} B_{m,j}^{(1)}, b_j = \sum_{m=M}^{\infty} B_{m,j}^{(2)}$, then $a_j \leq b_j$. Because $\boldsymbol{B}_{:,j_1}^{(k)} \preceq \boldsymbol{B}_{:,j_2}^{(k)}$ when $j_1 < j_2$, then

$$a_0 \leq a_1 \leq a_2 \leq ... \tag{23}$$

and

$$b_0 \leq b_1 \leq b_2 \leq ... \tag{24}$$

By the notations, we have that

$$\sum_{j=0}^{\infty} \left(\sum_{m=M}^{\infty} B_{m,j}^{(1)}\right) v_j \leq \sum_{j=0}^{\infty} \left(\sum_{m=M}^{\infty} B_{m,j}^{(2)}\right) w_j$$

$$\iff \sum_{j=0}^{\infty} a_j v_j \leq \sum_{j=0}^{\infty} b_j w_j$$

$$\iff \sum_{j=0}^{n_i} a_j v_j \leq \sum_{j=0}^{n_i} b_j w_j$$

The last equivalence is derived from that $v_j = w_j = 0$ when $j > n_i$. Consider the left part of last inequality. Employing $\boldsymbol{v} \preceq \boldsymbol{w} \Leftrightarrow \sum_{j=J}^{n_i} v_j \leq \sum_{j=J}^{n_i} w_j$ and Eq.24, the following inequality can be derived

$$\sum_{j=0}^{n_i} a_j v_j \leq \sum_{j=0}^{n_i} b_j v_j = b_0 v_0 + \sum_{j=1}^{n_i} b_j v_j \leq b_0 \left(\sum_{j=0}^{n_i} w_j - \sum_{j=1}^{n_i} v_j\right) + \sum_{j=1}^{n_i} b_j v_j$$

$$\leq b_0 w_0 + b_1 \left(\sum_{j=1}^{n_i} w_j - \sum_{j=1}^{n_i} v_j\right) + \sum_{j=1}^{n_i} b_j v_j = b_0 w_0 + b_1 w_1 + b_1 \left(\sum_{j=2}^{n_i} w_j - \sum_{j=2}^{n_i} v_j\right) + \sum_{j=2}^{n_i} b_j v_j$$

$$\leq \sum_{j=0}^{1} b_j w_j + b_2 \left(\sum_{j=2}^{n_i} w_j - \sum_{j=2}^{n_i} v_j\right) + \sum_{j=2}^{n_i} b_j v_j = \sum_{j=0}^{2} b_j w_j + b_2 \left(\sum_{j=3}^{n_i} w_j - \sum_{j=3}^{n_i} v_j\right) + \sum_{j=3}^{n_i} b_j v_j$$

$$...$$

$$\leq \sum_{j=0}^{n_i-1} b_j w_j + b_{n_i} (w_{n_i} - v_{n_i}) + b_{n_i} v_{n_i} = \sum_{j=0}^{n_i} b_j w_j$$

Therefore the left part is less than the right part, i.e. Eq.22 is established and the theorem is proved. $\square$

### A.1.7 THE PROOF OF PROPOSITION 4

*Proof.* Consider any input region $\mathbb{D}$. Let $\mathbb{D}'$ is the corresponding output region, i.e. $\mathbb{D}' = \boldsymbol{A}(\mathbb{D})$. By Proposition 1, $\mathrm{Sd}(\mathbb{D}') \leq \min\{\mathrm{Sd}(\mathbb{D}), k\}$. Because

$$\mathrm{Hist}(\{\min\{\mathrm{Sd}(\mathbb{D}), k\}|\ \mathbb{D} \text{ is any input region}\}) = \mathrm{cl}_k(\boldsymbol{v})$$

then $w \preceq \text{cl}_k(v)$. Suppose $v \in \mathbb{R}^{n+1}$, i,e, $v_i = 0$ when $i > n$, it is easy to verify that $\text{cl}_k(v) = M_{n,k}v$, where

$$M_{n,k} \in \mathbb{R}^{(k+1)\times(n+1)}, (M)_{i,j} = \delta_{i,\min(j,n+1)}$$

$\square$

### A.1.8 The proof of Proposition 5

Before the proof, we show the following lemma.

**Lemma 1.** *Suppose $\gamma_{n,n'}$ satisfies the recursion formula in Theorem 1 and the initial value satisfies that $|\gamma_{n_1,n_2}|_1 = \sum_{s=0}^{n_1} \binom{n_2}{s}$, then any $\gamma_{n,n'}$ satisfies $|\gamma_{n,n'}|_1 = \sum_{s=0}^{n} \binom{n'}{s}$.*

*Proof.* Since $\gamma_{n,n'} = \gamma_{n-1,n'-1} + \text{dm}(\gamma_{n,n'-1})$ and $\text{dm}(\cdot)$ does not change $|\cdot|_1$, we have $|\gamma_{n,n'}|_1 = |\gamma_{n-1,n'-1}|_1 + |\gamma_{n,n'-1}|_1$. By the assumption, suppose that $|\gamma_{n_1,n_2}|_1 = \sum_{s=0}^{n_1} \binom{n_2}{s}$ is established when $n_1 \leq n$ and $n_2 < n'$. Then

$$\begin{aligned}
|\gamma_{n,n'}|_1 &= \sum_{s=0}^{n-1} \binom{n'-1}{s} + \sum_{s=0}^{n} \binom{n'-1}{s} \\
&= \sum_{s=1}^{n} (C_s^{n'-1} + C_{s-1}^{n'-1}) + C_0^{n'-1} \\
&= \sum_{s=1}^{n} C_s^{n'} + C_0^{n'} \\
&= \sum_{s=0}^{n} \binom{n'}{s}
\end{aligned}$$

Therefore any $\gamma_{n,n'}$ satisfies the formula. $\square$

It is easy to verify that $|\gamma_{1,n}|_1 = n + 1 = \sum_{s=0}^{1} \binom{n}{s}$ and $|\gamma_{2,1}|_1 = 2 = \sum_{s=0}^{2} \binom{1}{s}$. By Lemma 1, $\gamma_{n,n'}$ proposed by us satisfies that $|\gamma_{n,n'}|_1 = \binom{n'}{s}$. Next we prove Proposition 5.

*Proof.* Denote the maxout layer by $h$, according to the proof of Theorem 10 in Serra et al. (2018), one $k$-rank maxout layer with $n_l$ output nodes corresponds to divide one region by $\frac{k(k-1)}{2}n_l$ hyperplanes. Suppose $R$ is one input region with $\text{Sd}(R) = n'$ and partitioned into $p$ sub-regions $\{r_1, ..., r_p\}$. Since one $d$-dimension space is at most partitioned into $\sum_{s=0}^{d} \binom{m}{s}$ sub-regions by $m$ hyperplanes, then by Lemma 1

$$p \leq \sum_{s=0}^{n'} \binom{\frac{k(k-1)n_l}{2}}{s} = |\gamma_{n',c}|_1$$

For any sub-region $p_i$, $\text{Sd}(p_i) = n'$. Since $h$ is equivalent to an affine transform in $p_i$ with matrix $A$ of rank $n_A$, we have that $\text{Sd}(h(p_i)) \leq \min\{n_A, n'\}$ by Proposition 1. Another fact is that $n_A \leq n_l$. Therefore $\text{Sd}(h(p_i)) \leq \min\{n_l, n'\}$. That is to say, $R$ with $n'$ space dimension is divided in to at most $|\gamma_{n',c}|_1$ sub-regions and the space dimension of each output sub-region is no larger than $\min\{n_l, n'\}$. If the space dimension histogram of input regions is $v \ (\in \mathbb{R}^n)$, then it is easy to verify that after the partition by $h$, the histogram of sub-regions $v'$ satisfies that $v' \preceq \text{diag}\{|\gamma_{0,c}|_1, \ldots, |\gamma_{n,c}|_1\} v$. In addition, $h$ change their space dimension. Thus

$$w \preceq \text{cl}_{n_l}(v') \preceq \text{cl}_{n_l}(\text{diag}\{|\gamma_{0,c}|_1, \ldots, |\gamma_{n,c}|_1\} v).$$

Let $C = \text{diag}\{|\gamma_{0,c}|_1, \ldots, |\gamma_{n,c}|_1\}$, then $\text{cl}_{n_l}(\text{diag}\{|\gamma_{0,c}|_1, \ldots, |\gamma_{n,c}|_1\} v) = M_{n,n_l}Cv$. $\square$

### A.1.9 THE PROOF OF PROPOSITION 6

*Proof.* When the skip connection is not added, $\boldsymbol{w} \preceq \boldsymbol{Bv}$. Let $\boldsymbol{v} = \boldsymbol{e}^n$, i.e. the the number of the input regions is one. Suppose the region is $R \subseteq \mathbb{R}^m$ and $R$ is divided into $p$ sub-regions. Apparently $p \leq |\boldsymbol{Be}^n|$ and $\mathrm{Sd}(R) = \mathrm{Sd}(r) = n \leq m$ where $r$ is one of the sub-regions. Suppose the part of network which is from the $i^{\text{th}}$ layer to the $j^{\text{th}}$ layer is equivalent to an affine transform with matrix $\boldsymbol{C}$. Let $r' = \boldsymbol{C}(r)$. When the skip connection is added, the output of $r$ is $\begin{bmatrix} \boldsymbol{C} \\ \boldsymbol{I} \end{bmatrix} (r)$ denoted by $r''$.

Since $\mathrm{rank}\left( \begin{bmatrix} \boldsymbol{C} \\ \boldsymbol{I} \end{bmatrix} \right) = m$, $\mathrm{Sd}(r'') \leq \min\{m, \mathrm{Sd}(r)\} = n$. This implies that the space dimension of $r'$ may be enhanced to $n$. Therefore the space dimension histogram of $p$ sub-regions $\boldsymbol{w}_R$ satisfies that

$$\boldsymbol{w}_R \preceq |Be^n|_1 \boldsymbol{e}^n. \tag{25}$$

For any input region with $n$ space dimension, Eq.25 is always established. Thus, when the space dimension histogram of input regions is $\boldsymbol{v}$ the histogram of output regions satisfies

$$\boldsymbol{w} \preceq \sum_R \boldsymbol{w}_R \preceq \sum_R |\boldsymbol{Be}^{\mathrm{Sd}(R)}|_1 \boldsymbol{e}^{\mathrm{Sd}(R)} = \sum_n v_n |\boldsymbol{Be}^n|_1 \boldsymbol{e}^n. \tag{26}$$

Let $\boldsymbol{C} = \mathrm{diag}\{|\boldsymbol{Be}^0|_1, |\boldsymbol{Be}^2|_1, ..., |\boldsymbol{Be}^n|_1, ...\}$, then $\sum_n v_n |\boldsymbol{Be}^n|_1 \boldsymbol{e}^n = \boldsymbol{Cv}$. $\qquad\square$

### A.1.10 THE PROOF OF PROPOSITION 7

*Proof.* Any residual structure can be regarded as the composition of one skip connection and an linear transform. For any input region $r \subseteq \mathbb{R}^m$ partitioned by the network, the residual structure part has the following form.

$$\boldsymbol{y} = \mathrm{Res}(\boldsymbol{x}) = [\boldsymbol{I} \quad \boldsymbol{I}] \begin{bmatrix} \boldsymbol{C} \\ \boldsymbol{I} \end{bmatrix} \boldsymbol{x}, \boldsymbol{x} \in r$$

By Proposition 6, the space dimension histogram of output regions $\begin{bmatrix} \boldsymbol{C} \\ \boldsymbol{I} \end{bmatrix} (r)$ denoted by $\boldsymbol{w}$ satisfies that

$$\boldsymbol{w} \preceq \sum v_n |\boldsymbol{Be}^n|_1 \boldsymbol{e}^n$$

Since $\mathrm{rank}([\boldsymbol{I} \quad \boldsymbol{I}]) = m \geq \mathrm{Sd}(r)$, according to Proposition 1 the linear transform will not change the histogram. $\qquad\square$

### A.1.11 PROPOSITION 8 AND ITS PROOF

**Proposition 8.** *For an MLP, let $f_m$ represent the first $m$ layers $(1 \leq m \leq l)$, i.e. $f_m(\boldsymbol{x})$ is the output of the $m^{th}$ layer, and $F_{l+1}(\boldsymbol{z}) = \sigma(\boldsymbol{W}^{(l+1)}\boldsymbol{z} + \boldsymbol{b}^{(l+1)})$ representing the $(l+1)^{th}$ layer in the MLP. Consider another network layer,*

$$G_{l+1}(\boldsymbol{z}, \boldsymbol{y}) = \sigma\left( \boldsymbol{W}^{(l+1,\prime)} \begin{bmatrix} \boldsymbol{z} \\ \boldsymbol{y} \end{bmatrix} + \boldsymbol{b}^{(l+1,\prime)} \right)$$

*where $\boldsymbol{z} = f_l(\boldsymbol{x}) \in \mathbb{R}^{n_l}, \boldsymbol{y} = f_m(\boldsymbol{x}) \in \mathbb{R}^{n_m}, 1 < m < l$. Then, given a specific $F_{l+1}$, there exists a $G_{l+1}$ such that the the total number of regions partitioned by $G_{l+1}$ is no more less than that by $F_{l+1}$.*

*Proof.* For a connected set $R \subseteq$ input space, $f_l(R)$ and $f_m(R)$ are still connected sets. For each hyperplane $H_i$ represented by $\sum_{j=1}^{n_l} a_{i,j} z_j + b_i = 0$ in $F_{l+1}$, design corresponding hyperplane $H'_i$, $\sum_{j=1}^{n_l} a_{i,j} z_j + \sum_{j=1}^{n_m} c_{i,j} y_j + b_i = 0$, in $G_{l+1}$, where $c_{i,j} = 0$. Suppose $H_i$ crosses $\{R_1, R_2, ..., R_{k_i}\}$ and take $k_i$ intersections $\{\boldsymbol{p}_1, \boldsymbol{p}_2, ..., \boldsymbol{p}_{k_i}\}$ where $\boldsymbol{p}_j$ is a interior point in $R_j (1 \leq j \leq k_i)$. Let $f_l(\boldsymbol{x}_j) = \boldsymbol{p}_j$, then $(f_l(\boldsymbol{x}_j), (f_m(\boldsymbol{x}_j)) \in H'_i$ and it is also a interior point in $R'_j$ which is the same as $R_j$ when cutting last $n_m$ dimensions. So $H'_i$ crosses at least $k_i$ regions which indicates the total number of regions partitioned by $G_{l+1}$ is no more less than that by $F_{l+1}$. $\qquad\square$

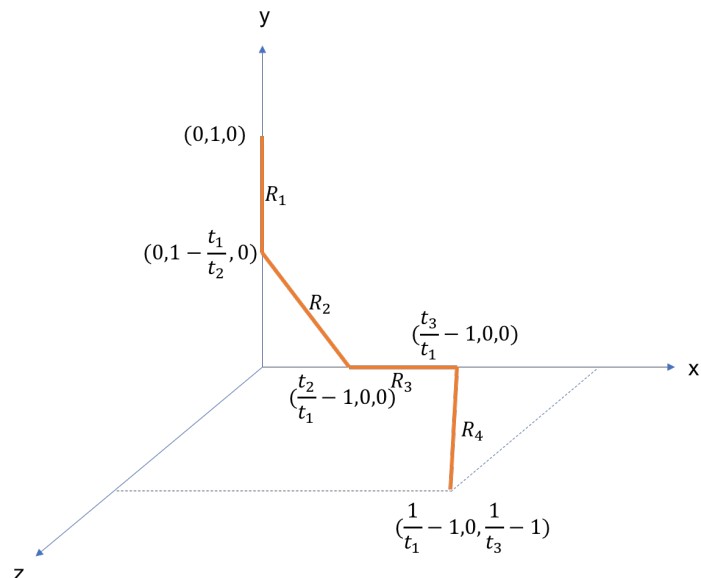

Figure 3: The four output regions in $\mathbb{R}^3$

### A.1.12 Proposition 9 and its proof

**Proposition 9.** *For a three-layer MLP which has two hidden layers, i.e.*

$$f(x) = \boldsymbol{W}^{(3)}\sigma(\boldsymbol{W}^{(2)}\sigma(\boldsymbol{W}^{(1)}\boldsymbol{x} + \boldsymbol{b}_1) + \boldsymbol{b}^{(2)}) + \boldsymbol{b}^{(3)}, \tag{27}$$

*suppose that $n_0 = 1$ and every segment partitioned by the first layer keeps their space dimension, i.e. their output of the first layer is still segment. If the input is concatenated to the output of the first layer (just like a skip connection), then no matter what the parameters in the first layer are, the practical maximum number of linear regions is $(n_1 + 1)(n_2 + 1)$. Specifically, without the special structure (just like an MLP) and assume that $n_1 = 3$, there exists some parameters in the first layer such that the practical maximum number is no more than $(n_1 + 1)(n_2 + 1) - n_2$.*

*Proof.* Denote $h_1(\boldsymbol{x}) = \sigma(\boldsymbol{W}^{(1)}\boldsymbol{x} + \boldsymbol{b}^{(1)})$. The first layer divides input space into $n_1 + 1$ regions $\{r_1, r_2, ...r_{n_1+1}\}$, then $h_1(r_i) \subseteq \mathbb{R}$. Because of the skip connection, the dimension of hyperplanes in the second layer is $n_1$. It is apparent that for any $n_1 + 1$ points there always exits a hyperplane containing them. Thus, for any interior point $x_i \in r_i (1 \leq i \leq n_1 + 1)$, there exists a hyperplane contaning them and therefore crossing $n_1 + 1$ regions. We have $n_2$ hyperplanes in the second layer, which shows the number of linear regions is $(n_1 + 1)(n_2 + 1)$. And obviously the maximum number is not larger than it (see Theorem 7 in Serra et al. (2018))

As for the special case, take $[0, 1]$ as input space without loss of generality. Let

$$\mathbb{W}_1 = \begin{bmatrix} \frac{1}{t_1} \\ -\frac{1}{t_2} \\ \frac{1}{t_3} \end{bmatrix}, \mathbb{1} = \begin{bmatrix} -1 \\ 1 \\ -1 \end{bmatrix} \tag{28}$$

Then we get four output regions in $\mathbb{R}^3$ shown in Figure 3. We can see that $R_1, R_2$ and $R_3$ are on the same plane, i.e. the x-y plane. For any interior points $\boldsymbol{p}_i \in R_i, i = 1, 2, 3$, the only hyperplane crossing them is x-y plane which is not able to divide $R_1, R_2$ and $R_3$. So the practical maximum number is no more than $(n_1 + 1)(n_2 + 1) - n_2$. □

## A.2 EXAMPLES

### A.2.1 AN EXAMPLE OF $\gamma_{n,n'}^{\text{new}}$ AND $\gamma_{n,n'}^{\text{serra}}$

Here $n' = 6$. Then $\gamma_{n,n'}^{\text{ours}}$ and $\gamma_{n,n'}^{\text{serra}}$ are shown as follows.

$$
\begin{bmatrix}
0 & 0 & 0 & 0 & 0 & 0 & 1 \\
0 & 0 & 0 & 0 & 1 & 6 & 6 \\
0 & 0 & 1 & 4 & 14 & 15 & 15 \\
0 & 2 & 5 & 16 & 20 & 20 & 20 \\
0 & 2 & 9 & 15 & 15 & 15 & 15 \\
0 & 2 & 6 & 6 & 6 & 6 & 6 \\
1 & 1 & 1 & 1 & 1 & 1 & 1
\end{bmatrix}
\quad
\begin{bmatrix}
0 & 0 & 0 & 0 & 0 & 0 & 1 \\
0 & 0 & 0 & 0 & 0 & 6 & 6 \\
0 & 0 & 0 & 0 & 15 & 15 & 15 \\
0 & 0 & 0 & 20 & 20 & 20 & 20 \\
0 & 0 & 15 & 15 & 15 & 15 & 15 \\
0 & 6 & 6 & 6 & 6 & 6 & 6 \\
1 & 1 & 1 & 1 & 1 & 1 & 1
\end{bmatrix}
\tag{29}
$$
$$\gamma_{\cdot,6}^{\text{ours}} \qquad\qquad\qquad\qquad \gamma_{\cdot,6}^{\text{serra}}$$

### A.2.2 AN EXAMPLE OF UPPER BOUND COMPUTATION FOR U-NET

We take the U-net in Appendix A.3.1 as example. Firstly, we use matrices to represent all layers except skip connections. When computing upper bound convolutional layers are regarded as fully-connected layers denoted by $C_i$. Suppose pooling layers are average pooling layers and unpooling ones are filling-zero ones. They are denoted by $P_i$ and $U_i$. Here, the subscript $i$ means the order in the network. Then according to Proposition 3 and Proposition 4 we have

$$C_1 = B_{2304}, C_2 = B_{1152}, C_3 = B_{576}, C_4 = B_{288},$$
$$C_5 = B_{144}, C_6 = B_{288}, C_7 = B_{576}, C_8 = B_{2304}$$
$$P_1 = M_{2304,576}, P_2 = M_{1152,288}, P_3 = M_{576,144},$$
$$U_1 = M_{144,144} = I_{144}, U_2 = M_{288,288} = I_{288}, U_3 = M_{576,576} = I_{576}.$$

where $B$, $M$ are defined by Proposition 3. By Proposition 6, the upper bound $N$ is computed as follows.

$$S_3' = U_1 C_5 M_{288,144} C_4 M_{144,288} P_3 \in \mathbb{R}^{144 \times 576}$$
$$S_3 = \text{diag}\{|S_3' e^0|_1, |S_3' e^2|_1, \ldots, |S_3' e^{576}|_1,\} \in \mathbb{R}^{576 \times 576}$$
$$S_2' = U_2 C_6 M_{576,288} S_3 C_3 M_{288,576} P_2 \in \mathbb{R}^{288,1152}$$
$$S_2 = \text{diag}\{|S_2' e^0|_1, |S_2' e^2|_1, \ldots, |S_2' e^{1152}|_1,\} \in \mathbb{R}^{1152 \times 1152}$$
$$S_1' = U_1 C_7 M_{1152,576} S_2 C_2 M_{576,1152} P_1 \in \mathbb{R}^{576,2304}$$
$$S_1 = \text{diag}\{|S_1' e^0|_1, |S_1' e^2|_1, \ldots, |S_1' e^{2304}|_1,\} \in \mathbb{R}^{1152 \times 1152}$$
$$N = |C_8 S_1 C_1 M_{576,2304} e^{576}|_1$$

### A.2.3 THE COMPARISON OF $B_n^{\text{ours}}$ AND $B_n^{\text{serra}}$

Here we consider $n = 6$. By Eq.29 and the definition of clipping function, we have

$$
\begin{bmatrix}
1 & 0 & 0 & 0 & 0 & 0 & 1 \\
0 & 7 & 0 & 0 & 1 & 6 & 6 \\
0 & 0 & 22 & 4 & 14 & 15 & 15 \\
0 & 0 & 0 & 38 & 20 & 20 & 20 \\
0 & 0 & 0 & 0 & 22 & 15 & 15 \\
0 & 0 & 0 & 0 & 0 & 7 & 6 \\
0 & 0 & 0 & 0 & 0 & 0 & 1
\end{bmatrix}
\quad
\begin{bmatrix}
1 & 0 & 0 & 0 & 0 & 0 & 1 \\
0 & 7 & 0 & 0 & 0 & 6 & 6 \\
0 & 0 & 22 & 0 & 15 & 15 & 15 \\
0 & 0 & 0 & 42 & 20 & 20 & 20 \\
0 & 0 & 0 & 0 & 22 & 15 & 15 \\
0 & 0 & 0 & 0 & 0 & 7 & 6 \\
0 & 0 & 0 & 0 & 0 & 0 & 1
\end{bmatrix}
$$
$$B_6^{\text{ours}} \qquad\qquad\qquad\qquad B_6^{\text{serra}}$$

### A.2.4 AN SIMPLE EXAMPLE OF IMPERFECT PARTITION

In this part, we use a simple example (in Figure 4) to illustrate imperfect partition. Consider a two-layer MLPs with $n_0 = 1, n_1 = 2$. The original input space is $\mathbb{R}$, i.e. the number axis or one line. The first layer partition the line into three parts (see Figure 4(a)) and the corresponding output regions in $\mathbb{R}^2$ are shown in Figure 4(b). In Figure 4(c), it is easy to observe that any hyperplane can not partition all three regions simultaneously.

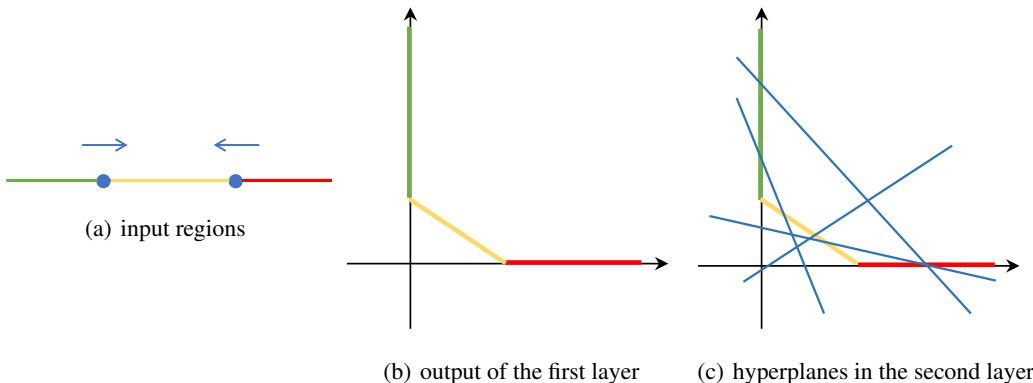

(a) input regions

(b) output of the first layer     (c) hyperplanes in the second layer

Figure 4: (a) The input region $\mathbb{R}$ is divided into three parts in three colors and activation directions of blue points are drawn above the line; (b) The output regions of the first layer; (c) The blue lines represent hyperplanes in the second layer

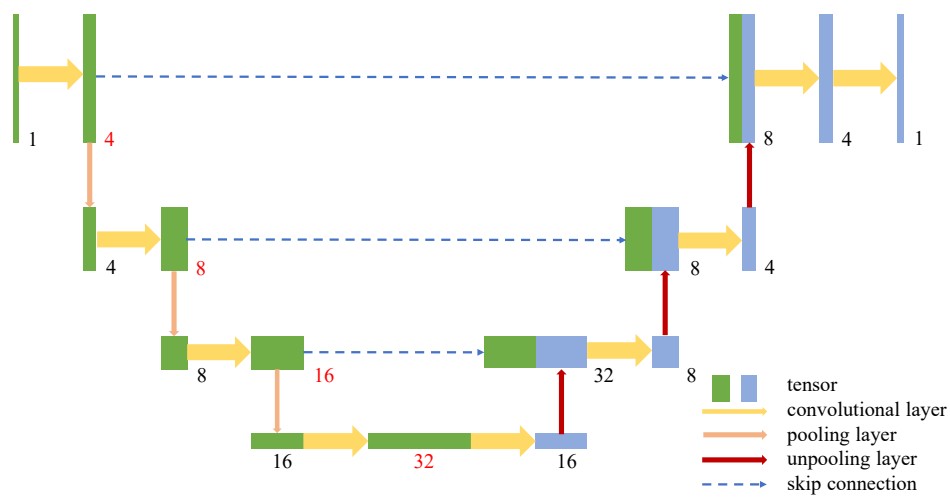

Figure 5: Network architecture with skip connections in No.1 of Table 1

## A.3 NETWORK ARCHITECTURES

### A.3.1 NETWORK ARCHITECTURES IN TABLE 1

We take the first setting in Table 1 as example. The numbers of "4-8-16-32" correspond to one in red color in Figure 5 and all the numbers represent the channel number of current tensor. For all setting in Table 1, the channel numbers of input and output are 1. The kernel size in every convolutional layer is $(3 \times 3)$ with stride $= 1$. We keep the size unchanged after the convolutional layer by padding zero. We use average-pooling as pooling layers and filling-zero as unpooling layers. The down-sampling rate and up-sampling rate are both 2. Except from the last convolutional layer, ReLU is added in other ones. The only differences in setting are channel numbers and the depth of down-sampling.

### A.3.2 NETWORK ARCHITECTURES IN TABLE 2

We also take the first setting in Table 2 as example. The numbers of "4-p16-p16-r16-r16-r16" correspond to one in red color in Figure 6. "p" means that before the convolutional layer, there exists a pooling layer. The last part of network is three fully-connected layers and ReLU is not added in the final layer. Other setting is the same as Appendix A.3.1.

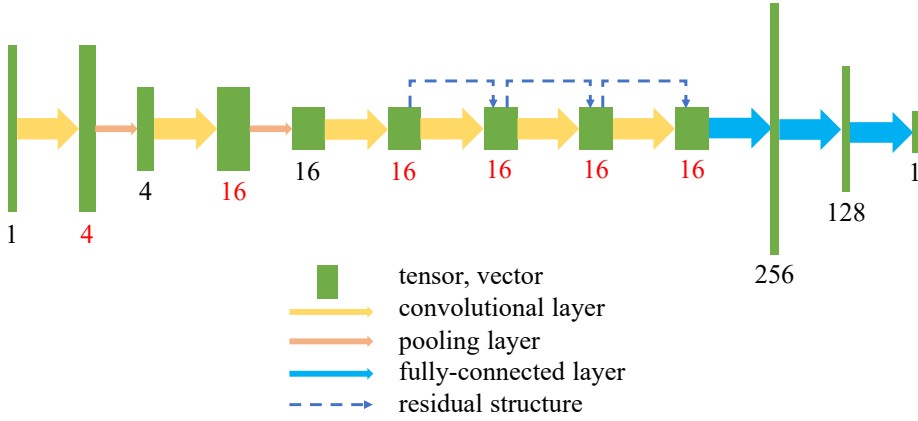

Figure 6: Network architecture with residual structures in No.1 of Table 2

