# OpenReview forum: "A General Computational Framework to Measure the Expressiveness of Complex Networks using a Tight Upper Bound of Linear Regions"
_ICLR.cc/2021/Conference — Reject_

### Official Review · AnonReviewer3 · 2020-10-26
**Weak Article on Number of Linear Regions in Deep ReLU Networks**

**Rating:** 3
**Confidence:** 4

**Review:**

This paper studies a method to give upper bounds on the total number of linear regions in a deep ReLU network. These bounds are tighter than those previously obtained and they apply not only to fully connected networks but also to networks with skip connections.

Strong Points:
1. Understanding the complexity of the function computed by a deep ReLU networks is of significant interest.
2. The authors state sharper bounds than previously available for this complexity, as measured by the number of linear regions.

Weak Points:
Many parts of the paper are not clear or imprecise:
1. Definition 1 is incompatible with the definition P(f) of the linear regions of f. For example, consider a ReLU networks with a single neuron. The partition R^{n_0} into linear regions simply divides the input space into two half-spaces. Thus, the regions D of this function are the closures of these half-spaces and hence are not disjoint (they share the same hyperplane as their boundary).

2. The definition of RL(n,n’) just before Definition 9 is not clear. Is there a bound on the number of neurons in the hidden layer? If one interprets “containing … one ReLU activation function” as having a single neuron, then it appears that (11) is a trivial histogram. If there can be any number of neurons, then in the definition 10 of when\gamma_{n,n’} satisfies the bound condition, the max of H_a(S_h) over h in RL(n,n’) appears to be a histogram whose entries are not well-defined since they are infinite. I am probably missing something but really didn’t understand what is meant here.

3. For a given ReLU network architecture, bounding the total number of linear regions in the worst case over all configurations of weights/biases is typically of no practical value (thought it is certainly an interesting problem in extremal combinatorics). The reason is that this number, even on average, is unbelievably large. This is true even in extremely simple networks. Take for instance a single hidden layer ReLU network in input dimension n_0, one hidden layer of size n_1 and output dimension 1. For generic configurations of the weights and biases, the number of linear regions grows like the sum of the binomial coefficients (n_1, j) as j goes from 0 to n_0 by Zaslavsky’s Theorem. So in the case when n_1 > n_0, for example, we get 2^{n_1} regions. This in no way reflects the fact that single layer ReLU networks often learn very simple functions. This is exemplified by the shocking numbers in Table 1 (where we see numbers like 10^{2956}).

Recommendation: I found that this paper was difficult to read. The results could be interesting from the point of view of a theoretical analysis of deep ReLU networks, so I encourage the authors to rewrite the body of the paper to increase the clarity and precision around the definitions of linear regions. Ultimately, I don’t think this paper is ready for publication.

---

### Official Review · AnonReviewer4 · 2020-10-28
**Missing transitions between the rather dense definitions and theorems**

**Rating:** 4
**Confidence:** 2

**Review:**

-------------- Overview of the paper -----------------

The number of regions can indicate the expressive power of neural networks. The paper provides a tighter upper bound of the regions number by providing a better choice of the \gamma parameter. The idea is built upon the work Hinz & Van de Geer, 2019, and the results is compared with Serra et al. (2018). The authors also study networks with pooling or skip connections.

-------------- Contribution and strength -----------------

I may not have a good sense of the contributions in the paper, since I am not very familiar with the related works. The main contribution is Theorem 1 and 2, which gives a better choice of \gamma. The experimental results then verifies the chosen \gamma parameters yields a tighter upper bound of the linear regions.

-------------- Questions and comments -----------------

The paper provides many background materials from Hinz&VandeGeer, 2019. Yet I feel like some intuitive explanations are missing, which makes it not very easy to follow closely. For example, what is the implication of the space dimension in definition 6?  Does it say something about the structure of a set or the size of a set? It should be helpful to provide some examples for finding the space dimension.

The better choice of \gamma in Theorem 1 and Theorem 2 also appear to be mysterious to me. Why we consider the downward-move function in definition 12? I am eager to look at an intuitive idea behind Theorem 1 and Theorem 2, rather a pure comparison between \gamma^{ours} v.s. \gamma^{serra}. By the way, \gamma^{ours} is not clearly defined in the paper.

Can the authors further elaborate on the extreme large ratio in Table 1?

-------------- Educated guess on the rating -----------------

I make an initial rating of the paper, yet it is quite open to discuss given my understanding of the topic. It seems the current paper lacks some transitions between the rather dense definitions and theorems in Section 2. This makes the paper hard to follow for diverse background.

I will reconsider the rating after the author response and seeing other reviews to reevaluate the value and soundness of the paper.

---

### Official Review · AnonReviewer1 · 2020-10-29
**Interesting topic, but poor presentation**

**Rating:** 4
**Confidence:** 2

**Review:**

This paper studies the counting of linear regions of a multi-layer ReLU network and gives an upper bound on the number of linear regions that is tighter than existing results. Networks with skip connections and pooling layers are also considered. The authors then compare their bounds for standard multi-layer networks and networks with skip connections/pooling layers and conclude that the latter has certain advantages in expressive power.

The paper is not written very clearly. It is very notion-heavy and lacks necessary explanations of the results. The definitions in this paper are pretty confusing and do not seem to be very rigorous. For example in Definition 3, a histogram v is defined as an infinite vector with non-negative integer entries and finite sum. However, in the discussion below the histogram is of length 5. More importantly, the definition of \gamma is very confusing: Definition 10 is not well-written, and I’m not quite sure if gamma can be any quantity satisfying the bound condition, or is it chosen more specifically.

The theoretical results are not presented very clearly either. It is not obvious to me what is the upper bound for the number of linear regions of a multi-layer ReLU network.

The discussion on pooling and skip connection is given without an explanation of the network structure. It is not clear to me how the matrix A and the skip connections appear in the network.

The comparison between the vanilla feed-forward network and the ones with skip connections/pooling layers may not be logically correct. Since the authors only compared the upper bounds of the number of linear regions, the experimental results do not necessarily demonstrate the advantage of networks with skip connections/pooling layers. It is possible that the bound for these types of networks are too loose.

From the technical aspect, it seems that the proof framework of this paper is mainly the same as previous works, except that a tighter bound of gamma is implemented. This makes the technical contribution of this paper limited.

For the above reasons, I would like to suggest that the authors should improve the presentation of the paper and explain the contributions more clearly and more convincingly.

---

### Official Review · AnonReviewer2 · 2020-11-01
**A General Computational Framework to Measure the Expressiveness of Complex Networks using a Tight Upper Bound of Linear Regions**

**Rating:** 4
**Confidence:** 3

**Review:**

Summary: This paper extends on the framework of matrix computation in Hinz & Van d Geer (2019) to give a tight upper bound for linear regions. In particular, the paper shows improvement over the bounds derived in Serra et al. 2018 and extends the bounds for more complex networks with skip and residual connections. The paper also shows why skip and residual connections can be beneficial by showing that they lead to networks with larger number of linear regions.

Pros:

1) The work of Hinz & Van d Geer (2019) can be seen as generalization of the results shown in Serra et al. 2018. Through some choice of lambda parameters used in the expression for bounds, this paper provides a tighter upper bound compared to Serral et al. 2018

2) The paper shows that in the presence of residual connections (ResNets), the number of linear regions is larger. This is understandable since there is no bottleneck effect in the case of ResNets as shown by Serra et al. 2018. The bottleneck effect leads to reduced dimensions if the dimension of the earlier layers are small. Having earlier layers with smaller dimensions can lead to decrease in the number of linear regions. However, in the case of residual networks, such an effect can not be observed. Thus we tend to have more linear regions in the case of residual connections.

Cons:

1) The paper needs significant improvement in the presentation. The parameter lambda, that is used in deriving the bounds, is not formally introduced, but the paper starts discussing this from the introduction.  It is difficult to appreciate the contributions or novelty, when the paper does not carefully explain the insights/ideas of the proposed formulation, nor the differences with prior work.

2) The results shown in this paper seem incremental in light of the work of Hinz & Van d Geer (2019).

---

### Decision · Program_Chairs · 2021-01-07
**Final Decision**

**Decision:**

Reject

**Comment:**

This paper studies the number of linear regions of a multi-layer ReLU network and gives a new upper bound. Reviewers concern about the writing and the results are incremental compared with previous results.